# Assessment of Tannin Tolerant Non-*Saccharomyces* Yeasts Isolated from *Miang* for Production of Health-Targeted Beverage Using *Miang* Processing Byproducts

**DOI:** 10.3390/jof9020165

**Published:** 2023-01-27

**Authors:** Pratthana Kodchasee, Nattanicha Pharin, Nakarin Suwannarach, Kridsada Unban, Chalermpong Saenjum, Apinun Kanpiengjai, Dipayan Sakar, Kalidas Shetty, Martin Zarnkow, Chartchai Khanongnucha

**Affiliations:** 1Division of Biotechnology, School of Agro-Industry, Faculty of Agro-Industry, Chiang Mai University, Chiang Mai 50100, Thailand; 2Research Center of Microbial Diversity and Sustainable Utilization, Chiang Mai University, Chiang Mai 50200, Thailand; 3Division of Food Science and Technology, School of Agro-Industry, Faculty of Agro-Industry, Chiang Mai University, Chiang Mai 50100, Thailand; 4Department of Pharmaceutical Science, Faculty of Pharmacy, Chiang Mai University, Chiang Mai 50200, Thailand; 5Division of Biochemistry and Biochemical Innovation, Department of Chemistry, Faculty of Science, Chiang Mai University, Chiang Mai 50200, Thailand; 6Department of Plant Sciences, North Dakota State University, Fargo, ND 58108, USA; 7Research Center Weihenstephan for Brewing and Food Quality, Technische Universität München, Alte Akademie 3, 85354 Freising, Germany; 8Research Center for Multidisciplinary Approaches to Miang, Chiang Mai University, Mueang, Chiang Mai 50200, Thailand

**Keywords:** non-*Saccharomyces* yeast, *Miang*, byproduct valorization, healthy beverage, fermentation

## Abstract

This research demonstrated an excellent potential approach for utilizing *Miang* fermentation broth (MF-broth), a liquid residual byproduct from the *Miang* fermentation process as a health-targeted beverage. One hundred and twenty yeast strains isolated from *Miang* samples were screened for their potential to ferment MF-broth and four isolates, P2, P3, P7 and P9 were selected, based on the characteristics of low alcoholic production, probiotic properties, and tannin tolerance. Based on a D1/D2 rDNA sequence analysis, P2 and P7 were identified to be *Wikerhamomyces anomalus*, while P3 and P9 were *Cyberlindnera rhodanensis*. Based on the production of unique volatile organic compounds (VOCs), *W. anomalus* P2 and *C. rhodanensis* P3 were selected for evaluation of MF-broth fermentation via the single culture fermentation (SF) and co-fermentation (CF) in combination with *Saccharomyces cerevisiae* TISTR 5088. All selected yeasts showed a capability for growth with 6 to 7 log CFU/mL and the average pH value range of 3.91–4.09. The ethanol content of the fermented MF-broth ranged between 11.56 ± 0.00 and 24.91 ± 0.01 g/L after 120 h fermentation, which is categorized as a low alcoholic beverage. Acetic, citric, glucuronic, lactic, succinic, oxalic and gallic acids slightly increased from initial levels in MF-broth, whereas the bioactive compounds and antioxidant activity were retained. The fermented MF-broth showed distinct VOCs profiles between the yeast groups. High titer of isoamyl alcohol was found in all treatments fermented with *S. cerevisiae* TISTR 5088 and *W. anomalus* P2. Meanwhile, *C. rhodanensis* P3 fermented products showed a higher quantity of ester groups, ethyl acetate and isoamyl acetate in both SF and CF. The results of this study confirmed the high possibilities of utilizing MF-broth residual byproduct in for development of health-targeted beverages using the selected non-*Saccharomyces* yeast.

## 1. Introduction

*Miang* is an ethnic fermented tea (*Camellia sinensis* var. *assamica*) from northern Thailand, which is fermented naturally by mixed microorganisms. This fermented tea is a unique traditional food product known as chewing tea or eating tea. It is believed to provide health benefits due to its specific bioactive compounds with antioxidant and antimicrobial activities [1]. The natural microbial communities involved in biotransformation processes, including lactic acid bacteria, yeast, *Bacillus* spp. and filamentous fungi, have been proposed as the key microbes in the successive process of *Miang* fermentation [2,3,4]. These biotransformation processes cause the unique strong aromatic flavor and taste of this fermented food. *Miang* fermentation process can be divided into two types, the non-filamentous fungi growth-based process (NFP) and filamentous fungi growth-based process (FFP). The NFP used the steamed young tea leaves as raw material in fermentation without the requirement for the growth of filamentous fungi, while the FFP used the steamed mature tea leaves as raw materials and required for the growth of filamentous fungi, as one important step involved in the fermentation process and product quality [1]. The FFP begins with the aerobic growth of filamentous fungus and subsequently leads to anaerobic fermentation. In local practice, anaerobic fermentation is performed either by a semi-submerged fermentation (SSF) or submerged fermentation (SMF) [3]. Once fermentation is completed, *Miang* products are harvested for the market, while the *Miang* fermentation broth (MF-broth) residual byproduct is not utilized for any other use and discharged as liquid waste byproduct. Kodchasee et al. [3] confirmed that some bioactive constituents, such as the organic acids, catechin and catechin-related compounds remained in this liquid waste after the *Miang* FFP. Yeast species are key microbes in *Miang* fermentation processes, yeast species have been suggested to contribute the unique aroma and flavor in *Miang* products. Some of the non-*Saccharomyces* yeast species have been reported as the microbial population found in *Miang* products, including *Candida ethanolica*, *C. boidinii*, *Cyberlindnera rhodanensis*, *Debaryomyces hansenii*, *Pichia manshurica*, and *Wikerhamomyces anomalus*. These non-*Saccharomyces* yeasts are tannin tolerant and have been suggested to be involved in the development of flavors and qualities of *Miang* [2]. Furthermore, it has been suggested that yeast species found in *Miang* fermentation processes are obtained from the natural environment of the *Miang* production area, such as non-*Saccharomyces* yeast from tea flowers that may be associated with *Miang* fermentation [2,5]. Some previous studies described the potential application in beverage production processes of using non-*Saccharomyces* yeasts *Cyberlindnera rhodanensis* and *Wikerhamomyces anomalus,* due to their ability to produce a special aroma. *W. anomalus* is recognized as a good producer of fruity esters, such as acetate esters, ethyl acetate, isoamyl acetate and 2-phenylethyl acetate [6]. As a result, these species were explored for flavor enhancement in several fermented beverage products [7]. Furthermore, the *Cyberlindnera* species has been also reported for producing high levels of acetate esters, isoamyl, ethyl and 2-phenylethyl acetate, especially *C. saturnus* (formerly *Williopsis saturnus*), *C. mrakii* (formerly *W. saturnus* var. *mrakii*) and *C. subsufficiens* [8,9].

Among the varieties of yeast, *Saccharomyces cerevisiae* has been recognized for playing an important role in the alcoholic beverage industry because of its ability to ferment sugar into ethanol and other metabolic products [10,11]. However, there are some limitations in fermentation using *S. cerevisiae,* such as low levels of beneficial flavors and odor development, therefore leading to for the need to find a non-*Saccharomyces* yeast species for improving the flavor and odor forming compounds during the fermentation process. Several non-*Saccharomyces* yeasts have gained high interest due to their capability for β-glucosidase production during alcoholic beverage fermentation and this enzyme is reported to be involved in aroma formation through the catalytic activity on the glycosidic bonds of monoterpenes or other glycosides, and subsequently release the aromatic components in fermented beverages, such as in wine and black tea [12]. *Dekkera bruxellensis*, *Hanseniaspora uvarum*, *Metschnikowia pulcherrima*, *Pichia kudriavzevii* and *Wickerhamomyces anomalus* (formerly known as *P. anomala*) have been used to improve aromatic properties of wine by their release of glucosidase [13,14]. The characteristics of non-*Saccharomyces* yeast is a lower alcohol production but it results in a higher aromatic compound production, compared to *Saccharomyces* yeast and therefore is attractive for the design of consumer-targeted healthy beverages [15]. Moreover, the use of non-*Saccharomyces* yeasts in many aspects of fermentation, such as in inhibiting the growth of unwanted microorganisms, detoxifying mycotoxin and increasing the bioactive compounds have been also reported [16]. Generally, non-*Saccharomyces* yeasts can be found in nature, such as in ripened fruit or flowers, as well as in beverages and fermented foods, such as kombucha, kefir and *Miang* (fermented tea) [2,17].

The fermented tea-beverages produced, based on either green or black tea substrates, such as kombucha and low-alcoholic beverages, have recently been increasing in popularity among the health-conscious consumers and the increased demand is leading to the development of beverages derived from tea substrates as well. Tea wines, the low-alcohol beverages made by adding the sugar or juice of tea, followed by fermentation with yeast, has also received of interest. The benefits of the tea derived compounds in these healthy beverages, such as antioxidant and antimicrobial properties and tea catechins have been important targets for value addition [18]. Furthermore, low alcoholic beverages also provide potential health benefits, such as lowering cholesterol and increasing high-density lipoprotein (HDL), which contributes to the prevention of heart disease and lowering the risks of alcohol-related illnesses [17,19]. The attractiveness of low-alcohol healthy drink products derived from tea supports the conceptual idea for the development of this potential health-targeted beverage using MF-broth, a waste byproduct from the *Miang* fermentation process, as the primary substrate. The rationale for this approach is both the design of a new healthy beverage and also the concurrent advancement of zero waste technology contributing to carbon sequestration through the utilization of MF-broth byproducts.

This study therefore describes the use of an MF-broth byproduct, together with the newly isolated indigenous species of non-*Saccharomyces* yeast from *Miang* in the development of a potential low-alcoholic healthy beverage. This product process has the potential to allow for the improvement of product quality through the enhancement of flavors and enrichment of bioactive constituents by the co-culturing of these yeast isolates. 

## 2. Materials and Methods

### 2.1. Strains, Media and Culture Conditions 

A total of 120 yeast strains used in this study and previously isolated from *Miang* samples by Kanpiengjai et al. [2] were maintained in 25% (*v/v*) glycerol and kept at −80 °C as a stock culture at the Laboratory of Microbial Resources Development and Enzyme Technology, Faculty of Agro-Industry, Chiang Mai University. The reference strain *S. cerevisiae* TISTR 5088 was purchased from the culture collection of the Thailand Institute of Scientific and Technological Research (TISTR). The yeast strains were routinely grown at 30 °C in yeast peptone dextrose (YPD) broth consisting of 1% (*w/v*) yeast extract, 2% (*w/v*) peptone and 2% (*w/v*) glucose.

### 2.2. Screening of the Yeasts for their Ethanol Producing Capability 

A single colony of each of the 110 yeast isolates was inoculated into a Durham tube containing 10 mL YPD broth and statically incubated at 30 °C and observed for gas formation following 24 h and 48 h cultivation. The yeast isolates capable of gas formation were selected and further investigated for glucose consumption and yield of ethanol production. The ethanol and glucose concentrations from the selected yeast isolates growing in the culture broth were analyzed by high-performance liquid chromatography (HPLC), as described by Kodchasee et al. [3].

### 2.3. Probiotic Characterization and Tannin-Tolerance Test

The selected ethanol producing yeasts were investigated for their probiotic characteristics, which included the ability to grow at 37 °C, the resistance to grow in bile salt and low pH conditions, hemolytic activity and the adhesion capacity of cells [20]. Briefly, yeast cells were incubated in a simulated gastric juice containing 0.35% (*w/v*) pepsin at pH 2.0 and 3.0, and 0.3% (*w/v*) bile salt in phosphate buffered saline (PBS buffer) at pH 7 (PBS buffer pH 7 as control). The treated suspension was incubated at 37 °C for 3 h. Viable cell counts were determined by plating on yeast malt (YM) agar at 37 °C for 24 h, and the survival percentage was calculated as follows: survival (%) = [final (logCFU/mL)/control (logCFU/mL)] × 100. Hemolytic activity was determined by inoculating the yeast strain on a blood agar plate containing 5% defibrinated sheep blood and incubated at 37 °C for 72 h. The development of a clear zone of hydrolysis surrounding the colonies was observed and classified [21]. The cell surface hydrophobicity was tested in toluene (a nonpolar solvent). The yeast suspension was prepared in PBS to an optical density of 600 nm (OD600) = 1 (A0). Then, the volume of toluene was added into the yeast cell suspension at a ratio of 1:1 and the two-phase system was mixed for 5 min. Following 1 h of incubation at 37 °C, OD600 of the cell suspension was measured (A1) and the microbial adhesion to solvents (MATS) percentage was calculated as follows: MATS (%) = [(A0 − A1)/A0] × 100. Isolates with MATS above 50% were considered as hydrophobic [22]. The tannin-tolerance of the selected yeast isolates was also investigated on YM agar containing tannin, according to the method described by Kanpiengjai et al. [2]. A single colony of yeast isolate was picked up and spiked on YM agar with 10, 30 and 50 g/L tannin. The growth of the yeast isolates was observed after incubation at 30 °C for 3 days. The characteristics of low alcoholic producing properties, probiotic properties and tannin tolerance were used as the criteria for the selection of the proper yeast strain used in the healthy beverage fermentation using *Miang* fermentation broth residual byproducts.

### 2.4. Molecular Identification and Carbon Assimilation Profiles

The selected yeasts were identified, based on the morphological characteristics, biochemical tests, and the presence of D1/D2 region of the 26S rRNA gene. Extraction of genomic DNA was performed as described by Kanpiengjai et al. [2] and the PCR was performed using two universal primer pairs NL1 (5’-GCATATCAATAA GCGGAGGAAAAG-3’) and NL4 (5’-GGTCCGTGTTTCAAGACGG-3’). The 26S rRNA gene were sequenced and analyzed using the BLAST algorithm of GenBank (http://www.ncbi.nlm.nih.gov/blast, accessed on 25 November 2022) and were deposited under the accession numbers OL63634 (P2 strain), OL636341 (P3 strain), OL636342 (P7 strain) and ON460286 (P9 strain). Carbon assimilation was performed with API ID 32C strips (Biomérieux, Craponne, France) and carbon fermentation was analyzed and compared, according to a standard yeast identification method [23]. The samples were evaluated visually for turbidity in wells, differentiating the positive (+), negative (−) and weak (w) growths.

### 2.5. Analysis of the Volatile Compounds

The selected yeast strains were investigated for their ability to produce volatile compounds (VOCs) during cultivation on YM slants at 30 °C in the dark, following the method described by Suwannarach et al. [24]. The VOCs in the air space above each yeast colony were captured for 45 min at 25 ± 2 °C using a solid-phase microextraction (SPME) fiber, composed of 50/30 divinylbenzene/carboxen on polydimethyl-siloxane on a stable flex fiber (Supelco, Bellefonte, PA, USA). The adsorbed fiber was placed into the splitless injection port of a gas chromatograph (Agilent Technologies Inc, Palo Alto, CA, USA) equipped with a DB-WAX capillary column (0.25 mm × 30 m I.D × 0.25 μm film thickness, Supelco, Bellefonte, PA, USA). The column temperature was set to 40 °C for 2 min, then to 200 °C for 5 min. At an initial column head pressure of 60 kPa, the carrier gas was ultra-high purity helium. Prior to trapping the volatiles, the fiber was cleaned at 250 °C for 57 min under a flow of helium gas. The adsorbed volatiles were introduced into the gas chromatography (GC) interfaced with a mass spectrometer over a 30-s injection duration (Agilent 5975C, Agilent Technologies Inc., Santa Clara, CA, USA). With ionization energy of 70 eV, the mass detector was operated in the electron impact mode. The GCMS solution was used to collect and process the data (Agilent). The volatiles were identified by comparison with a mass spectra library (NIST) and the RI of the pure standard compounds were confirmed by GC-MS (https://webbook.nist.gov/chemistry/ accessed on 8 March 2022) [24].

### 2.6. MF (Miang Fermentation) Broth Fermentation 

MF-broth, the remaining byproduct liquid waste from the submerged fermentation process in *Miang* production, was collected from the FFP *Miang* producing plant in Phrae province, Thailand. The starter cultures of two selected strains of non-*Saccharomyces* yeast species (*Wikerhamomyces anomalus* P2 and *Cyberlindnera rhodanensis* P3) and *S. cerevisiae* TISTR 5088 were prepared by cultivating the yeast strains in 100 mL YM broth at 30 °C for 24 h on a 150-rpm rotary shaker. The cells were harvested by centrifugation at 6000 rpm for 5 min at 4 °C, washed twice and re-suspended in a sterile solution of 0.85% NaCl (*w/v*) and used as the starter for fermentation. Glucose was added into the MF-broth to achieve the final concentration of 5% (*w/v*) glucose in 300 mL in a 500 mL flask and sterilized at 121 °C for 20 min. Seven fermentative groups consisting of three single microbe fermentations (SF) of *S. cerevisiae* TISTR 5088, *W. anomalus* P2 and *C. rhodanensis* P3 and four co-fermentations (CF) of *S. cerevisiae* TISTR 5088 + *W. anomalus* P2, *S. cerevisiae* TISTR 5088 + *C. rhodanensis* P3, *W. anomalus* P2 + *C. rhodanensis* P3, *S. cerevisiae* TISTR 5088 + *W. anomalus* P2 + *C. rhodanensis* P3 were conducted and a non-inoculated sterile broth was used as a control. The fermentation was carried out in a 500 mL Erlenmeyer flask with 5% (*v/v*) starter containing a final inoculated concentration of approximately 6.5 logCFU/mL *S. cerevisiae* and non-*Saccharomyces* yeast cultures at a 1:1 ratio in the *Miang* byproduct fermented broth and incubated at 121 °C for 5 days. Samples were taken at 0, 12, 24, 36, 48, 72, 96 and 120 h to determine the viable cell counts and pH. The bioactive compound in the fermented broth was determined by the method described by Kodchasee et al. [3]. The remaining sugars, organic acids and ethanol concentrations were analyzed by HPLC. Volatile compounds analyses (VOCs) were performed using headspace vials with the sample of the final fermentation time (120 h).

### 2.7. Enumeration of the Microbes and the pH Measurement 

The freshly fermented sample (1 mL) was transferred into 9 mL of 0.85% (*w/v*) NaCl solution and serially diluted with a 10-fold dilution. A volume of 10 μL of 10^−2^ to 10^−5^ dilutions were dropped on YM agar and incubated at 30 °C for 24 h. The viable cell counts of the yeast cultures were observed and presented as the log of a colony forming unit per milliliter (log CFU/mL). Measurement of the pH was performed by a Metrohm 744 pH meter with a glass electrode (Metrohm Co. Ltd., Herisa, Switzerland). 

### 2.8. Analysis of Glucose, Ethanol, Organic Acids, Catechins and Caffeine by HPLC

Glucose, ethanol, and organic acid were analyzed using HPLC, according to Kodchasee et al. [3]. Briefly, a solution of 10 μL of each sample was filtered through a 0.45 μm filter paper (Whatman Inc., Clifton, NJ, USA), and injected into the HPLC system (Agilent 1000 series, Agilent Technologies Inc., Palo Alto, CA, USA) equipped with a 150 × 7.80 mm Rezex ROA organic acid H+ (8%) column (Phenomenex, Torrance, CA, USA). A mobile phase of 0.005 M H_2_SO_4_ was used for the glucose and ethanol analyses via a refractive index (RI) detector with a flow rate of 0.6 mL/min and 60 °C temperature. The concentrations of glucose and ethanol were calculated, based on the area peak corresponding with their standard retention time (RT).

The organic acids analysis was carried out with the same column using an isocratic elution of 50% (A) 0.05 M H_2_SO_4_ and 50% (B), 2% acetonitrile with a flow rate of 0.5 mL/min; temperature 40 °C, with a UV detector at the wavelength of 210 nm. The concentration of organic acids was calculated, based on the retention time (RT) corresponding to standards (acetic, citric, formic, gallic, glucuronic, lactic, malic, oxalic, succinic, and tartaric acids). 

Catechins and caffeine were analyzed using a C18 column (250 × 4.6 mm, PhenomenexGemini, Torrance, CA, USA) with the HPLC conditions as follows; UV detector; mobile phase (A) 0.1% acetic acid in acetonitrile and mobile phase (B) 0.1% acetic acid in deionized (DI) water at a flow rate of 1.0 mL/min at 20 °C and injection volume of 10 μL. The concentrations of catechins and caffeine were calculated, based on their RT, compared with the standards that included epigallocatechin gallate (EGCG), epicatechin gallate (ECG), gallocatechin gallate (GCG), epigallocatechin (EGC), epicatechin (EC), catechins (C), gallocatechin (GC) and caffeine (Sigma-Aldrich, St. Louis, MO, USA) [25]. 

### 2.9. Determination of the Bioactive Compounds

The bioactive compounds containing total phenolic (TP), total tannin (TT) and total flavonoids (TF) contents were determined following the methods described by Kodchasee et al. [3]. TP was measured according to a modified Folin–Ciocalteu method. Briefly, 200 μL of fermented MF-broth was added into a glass test tube (12 × 100 mm) containing 200 μL of 2 M Folin–Ciocalteu reagent and mixed using a vortex mixer (Vortex Genie 2, Scientific Industries, Bohemia, NY, USA). Then, 1.8 mL of deionized water (DI water) was added, and the reaction mixture was incubated at ambient temperature for 3 min. Then, 400 μL of 10% (*w/v*) sodium carbonate was added. The volume was adjusted using DI water to 4 mL and incubated in the dark at ambient temperature for 1 h. The absorbance of the blue solution was measured at 725 nm using the spectrophotometer. The total phenolic content was determined from the calibration curve using gallic acid as the reference standard (0–200 mg/L of gallic acid). 

TT was determined by the Folin–Ciocalteu method and using polyvinylpolypyrrolidone (PVPP) to separate the tannins from other phenols. Briefly, 1 mL of the MF-broth sample was mixed with 1 mL of 10% (*w/v*) PVPP, vortexed and incubated at 4 °C for 15 min. Then, it was centrifuged at 3000× *g*, at 4 °C for 10 min. The remaining TP of the PVPP precipitated supernatant was measured with the Folin–Ciocalteu reagent and TT was estimated using the formula: TT = TP − PVPP precipitation. 

TF contents were determined using the aluminum chloride colorimetric method. Briefly, 250 μL *Miang* extract was mixed with 250 μL 10% (*w/v*) aluminum nitrate and 50 μL 1 M potassium acetate. Then, 1.65 mL 80% (*v/v*) methanol was added to the reaction mixture and incubated in the dark at ambient temperature for 40 min. The absorbance was measured at 415 nm. The content of the flavonoids was determined from the calibration curve using the quercetin equivalent as the reference standard (0–250 mg/L of quercetin equivalent).

Antioxidant activity was based on 1,1-diphenyl-2-picrylhydrazyl (DPPH) (Sigma-Aldrich, St. Louis, MO, USA) free radical assay [26]. Briefly, 100 μL of diluted sample in DI water (10 μL/mL) was mixed with 400 μL of 0.15 mM DPPH in 80% methanol and incubated in the dark at ambient temperature for 30 min. The absorbance was measured at 517 nm, the radical scavenging percentage was calculated against a blank using the following equation: inhibition (%) = (1 − (B/A)) × 100, where A is the absorbance of the mixture without sample, and B is the absorbance of the mixture containing the sample. 

### 2.10. β-Glucosidase Activity Assay

β-Glucosidase activity was assayed by measuring the amount of p-nitrophenol (pNP) liberated from *p*-nitrophenyl-β-D-glucoside (pNPG) as substrate [16]. First, 0.125 mL MF-broth was mixed with 0.125 mL of a 4 mM solution of pNPG in 0.1 M citrate–phosphate buffer (pH 5.0). The reaction mixture was incubated at 30 °C for 20 min and subsequently 0.250 mL of 50 mM sodium carbonate (Na_2_CO_3_) was added to stop the reaction. The absorbance of the final reaction mixture was measured at 405 nm by a spectrophotometer (Metertech SP-8001 UV/Visible Spectrophotometer, Metertech Inc., Taipei, Taiwan). The enzyme activity unit (U) was expressed as the ability to liberate 1 μmole pNP from the substrate solution/min. 

### 2.11. Statistical Analysis

All experiments were performed in triplicate and the data are represented as the mean ± standard deviation. Statistical analyses were conducted using IBM SPSS statistics 23.0 software package (SPSS Inc., Chicago, IL, USA) employing the Tukey multiple range tests at a significance level of *p* < 0.05. The principal component analysis (PCA), hierarchical cluster analysis and heat-maps were performed through GraphPad PRISM 9 (GraphPad Software, La Jolla, CA, USA) and SPSS software, respectively.

## 3. Results and Discussion

### 3.1. Screening of Yeast with Potential Characteristics

Among 120 yeast strains, 24 isolates showed a gas formation (Figure 1a) and the ethanol production in the culture broth were confirmed by HPLC, in comparison to the efficient ethanol producing yeast *Saccharomyces cerevisiae* TISTR 5088 (Figure 1b). All 24 gas producing yeasts showed a variety of both ethanol concentration and sugar consumption. Most of the sucrose (20 g/L) was consumed by the isolates P2, P3, P7 and P9, similar to *S. cerevisiae* TISTR 5088, but the lower ethanol yields were observed from the yeast isolates P2, P3, P7 and P9 at 2.5, 3.3, 3.1 and 2.7 g/g substrate, respectively, while the ethanol yield of 0.46 ± 0.02 g ethanol/g glucose (9.21 ± 0.35 g/L) was produced by *S. cerevisiae* TISTR 5088. Aligned with the purpose of this research, yeast isolates P2, P3, P7 and P9 were selected due to potential of high glucose consumption (>19.90 g/L) and low ethanol production, which were approximately 50% of the ethanol yield observed for the ethanol producer as *S. cerevisiae* TISTR 5088. 

The tolerance to both a low pH condition and bile salts were used to evaluate the probiotic characteristics. The selected isolates showed a high tolerance to pepsin at pH 2.0 after 3 h exposure, with a survival rate of 83.9 to 99.9% (Table 1), whereas the survival rate of bile salt was 66.6 to 91.4%. The highest probiotic potential was in isolate P2, which demonstrated the highest percentage of survival rate, both in bile salt and low pH conditions, followed by isolates P3 and P9, respectively. However, the strains are required to survive against the natural barriers of the gastro-intestinal tract, including body temperature, a low pH in the stomach and high bile acid concentration at the duodenum in the small intestine [22]. The hydrophobicity of the yeasts was assessed indirectly as the ability to interact with chloroform and the results showed the hydrophobicity index ranged from 31.14 to 52.31%. None of the yeast strains showed β-hemolysis after 5 days incubation (Figure 2a). Several reports have previously described the probiotic properties of non-*Saccharomyces* yeast species in beverage and food production [10,27,28,29,30]. This result shows the additional advantage to the potential health benefit using MF-broth. Although, *C. rhodanensis* were reported to be associated with clinical bovine mastitis [31], but the *C. rhodanensis* strains investigated in this study are negative for a hemolytic response. In addition to the edible source of the yeast isolate, the negative hemolytic test indicates the safety status for *C. rhodanensis* used in this study [32]. 

Tannin in *Miang* products is likely to influence the microbial growth and development throughout the fermentation process. Therefore, the tannin tolerance capability was investigated to predict whether the yeast could grow in MF-broth. Isolates P2, P3, P7 and P9 were cultured on YM agar with varied tannin concentrations of 10, 30 and 50 g/L, respectively. All isolates could grow in all tannin concentrations and formed clear zones around their colonies, indicating that they have a tannin degrading system initiated by the key enzyme as tannase (Figure 2b). The previous discovery of the cell associated tannase (CAT) with high capability in gallate and catechin biotransformation have been reported from yeasts isolated from *Miang*, *C. rhodanensis* A22.3 and *C. rhodanensis* A45.3 [33]. As a result, we expected that the yeast species obtained from the *Miang* sample would be able to grow in MF-broth containing tannins and increase the antioxidant activity because of the efficient tannin and catechin biotransformation caused by CATs.

### 3.2. Molecular Identification and Carbon Assimilation Profiles

Based on the result of the D1/D2 region nucleotide sequence analysis, the isolates P3 and P7 were identified to be *Cyberlindera rhodanensis*, while the isolates P2 and P9 were *Wickerhamomyces anomalus* (Figure 3). The results of the analysis of the biochemical characteristics of the yeasts are also summarized in Table 2. The carbon fermentation and assimilation analysis found that sucrose and glucose utilization were positive in all strains. Meanwhile, the assimilation of cellobiose, galactose, raffinose and xylose were different in *C. rhodanensis* P3 and P7 and *W. anomalous* P2 and P9. Interestingly, *C. rhodanensis* showed positive results for the assimilation of cellobiose, a known substrate of β-glucosidase, which is an enzyme involved in the aromatic compound formation in wine [13]. The several non-*Saccharomyces* yeasts isolated from fermented food were examined for their ability to improve wine fragrance during fermentation. From these studies, non-*Saccharomyces* yeasts were recognized for their low ethanol fermentation rate, in comparison to *S. cerevisiae*. *W. anomalus* has previously been investigated in the production of low alcohol with high levels of esters, such as in kiwi wine [34], muscat bailey wine [13], apple wine [35] and Baijiu [36]. The members of the genus *Cyberlindnera*, as well as *C. subsufficiens*, *C. mrakii*, *C. jadinii*, *C. fabianii* and *C. saturnus* [9,37,38,39] were studied on the generation of a fruity aroma in non-alcoholic beer, but *C. rhodanensis* has not been reported for use in fermented beverages. Therefore, two of the selected non-*Saccharomyces* yeasts *C. rhodanensis* isolated from a traditional fermented tea of FFP *Miang* are attractive to elucidate and develop for further applications, especially in the development of the value-added beverages from the waste byproduct produced during the *Miang* fermentation process.

The analysis of VOCs produced by *S. cerevisiae* TISTR 5088, *C. rhodanensis* P3 and P7 and *W. anomalous* P2 and P9 was performed by the headspace SPME-GC/MS technique that indicated that higher alcohols were the main VOCs of *S. cerevisiae*, while acetate esters were the main esters of the non-*Saccharomyces* yeast species. *S. cerevisiae* and *W. anomalus* P2 and P9 produced at least five VOCs on YM slant agar, while *C. rhodanensis* P3 and P7 produced 10 types of VOCs. The VOCs were analyzed by comparing their mass spectral characteristics to those in the NIST database (Figure 4). Isoamyl alcohol (63.54% of total area) was the most abundant aroma compound generated by *S. cerevisiae*, followed by 2-phenylmethyl ethanol and isoamyl acetate. Whereas acetic acid ethyl ester was the most abundant VOC in the culture of *W. anomalus* P2 and P9, followed by 1-butanol- 3-methyl acetate, 2-phennethyl acetate, 2-phenylmethyl ethanol and 2-phenylmethyl ethanol. 1-Butanol-3-methyl acetate was the most abundant VOC produced by *C. rhodanensis* P3 and P7, followed by acetic acid ethyl ester, 2-phennethyl acetate, 2-phenylmethyl ethanol and 2-phenylmethyl ethanol. Furthermore, 2-furyl methyl acetate, benzyl acetate and 2-propenoic acid were found in the fermented broth of *C. rhodanensis* P3 and P7 but not in *S. cerevisiae* or *W. anomalus* P2 and P9. However, esters are a type of volatile compound produced by yeast cells during alcoholic fermentation, and various studies have demonstrated that their presence is related to wine quality. Ethyl acetate and isoamyl acetate are the most important esters in wine as both are associated with fruity wine/apple-like sweet pear drop and sweet banana ripe flavors, respectively [40]. Since the strains of *C. rhodanensis* P3 and P7 and *W. anomalus* P2 and P9 produced significant amounts of ethyl acetate and isoamyl acetate, we decided to select the representative yeasts *C. rhodanensis* P3 and *W. anomalus* P2 for further studies, as both isolates produced the highest ester in each species. 

### 3.3. Yeast Growth Dynamic and pH Changes in MF (Miang Fermentation) Broth

The growth dynamics of *S. cerevisiae* TISTR 5088 and non-*Saccharomyces* yeast species in the respective single fermentation (SF) and co-fermentation (CF) of MF-broth fermentation are shown in Figure 5. The control MF-broth did not have any microbial growth, while the culture of non-*Saccharomyces* species P2 and P3 showed the increase of the viable yeast number log 6.80 ± 0.03 and log 6.61 ± 0.03 CFU/mL, respectively. In CF, the growth of the yeast population showed a similar trend in all treatments and reached the approximate level of log 6.60 ± 0.02 CFU/mL at 120 h fermentation. The pH of MF-broth slightly decreased after 120 h of fermentation in the range of 4.04 ± 0.03 to 3.88 ± 0.01, which is very close to the pH of the initial MF-broth (Figure 5). The MF-broth fermentation inoculated with non-*Saccharomyces* yeast species exhibited a lower growth than that of *S. cerevisiae* TISTR 5088, which might be due to the numerous parameters involved, such as the cultivability loss of non-*Saccharomyces* strains in alcoholic fermentation, the nitrogen limitation, low oxygen availability and inhibition by increased ethanol, as well as extrinsic factors, such as SO_2_ [41,42]. However, when *S. cerevisiae* and non-*Saccharomyces* yeasts were inoculated in a 1:1 ratio, the amount of yeast was slightly reduced in SF, indicating a mutual inhibition under the current conditions, especially *W. anomalus*, which has a reported killer activity against *S. cerevisiae*. To minimize the technological issues caused by sluggish or incomplete alcoholic fermentation, the compatibility of the selected killer *W. anomalus* strains with the primary microbial agents involved in wine production must be investigated throughout the selection stage [6,7,43]. 

### 3.4. Changes in the Sugar Consumption, Ethanol Production and Organic Acid Formation 

The glucose concentration in the original MF-broth was 3.56 ± 0.10 g/L, which occurred via the microbial degradation of tea leaf polysaccharides during *Miang* fermentation. This original sugar content is low for supporting yeast fermentation. The external source of glucose was added up to the final concentration of 5% (*w/v*) as the main carbon source for yeast growth and conversion to alcohol. The initial glucose concentration was at 53.21 ± 0.48 g/L. In SF, the glucose consumption ranged around 20–50 g/L at 120 h (Figure 5). *S. cerevisiae* TISTR 5088 showed a high glucose consumption, since a significant low residual concentration of glucose (0.19 ± 0.77 g/L) was detected (Table 3), followed by *C. rhodanensis* P3 and *W. anomalus* P2. In case of CF, a range of varied glucose concentrations (5 to 50 g/L) was observed, co-fermentation with *S. cerevisiae* TISTR 5088 clearly showed a higher glucose consumption than the SF of *C. rhodanensis* P3 and *W. anomalus* P2. The combination of *S. cerevisiae* TISTR 5088 with either *C. rhodanensis* P3 or *W. anomalus* P2, in CF, showed the almost complete consumption of glucose during the final period of fermentation. 

In SF, the ethanol production by *S. cerevisiae* TISTR 5088 was observed after 24 h and reached the maximum at 96 h with the highest ethanol concentrations of 23.67 g/L (Figure 5). Whereas the ethanol productions by *C. rhodanensis* P3 and *W. anomalus* P2 were observed after 48 h and showed a final ethanol concentration of 11.03 and 11.56 g/L, respectively, at 120 h (Table 3). The CF with *S. cerevisiae* TISTR 5088 showed a more rapid ethanol production after 12 h, both in response to *S. cerevisiae* TISTR 5088 + *W. anomalus* P2 and *S. cerevisiae* TISTR 5088 + *C. rhodanensis* P3, and an almost same level of ethanol production was observed (20.74 ± 0.21 and 20.14 ± 0.43 g/L, respectively), while *S. cerevisiae* TISTR 5088 + *W. anomalus* P2 + *C. rhodanensis* P3 produced the highest ethanol concentration at 24.91 ± 0.76 g/L. Moreover, CF of *W. anomalus* P2 + *C. rhodanensis* P3 produced a low amount of ethanol at only 2.92 ± 0.05 g/L. The greater ethanol production of *S. cerevisiae* TISTR 5088 in SF, when compared to *W. anomalus* P2 and *C. rhodanensis* P3 is probably caused from the weak metabolic response of non-*Saccharomyces* yeasts in consumption or the utilization of the nutrients in the culture broth. Furthermore, it has been reported that the low level of oxygen in the fermenting activity of this yeast leads to an increase in the cell biomass and a decrease in ethanol yield, which is a strategy that leads to the reduction of the ethanol level in wine produced by the co-culture with *S. cerevisiae* [44,45]. Therefore, in CF, the ethanol content might mainly come from *S. cerevisiae* TISTR 5088. Furthermore, the sugar content control in MF-broth fermented with non-*Saccharomyces* yeast species resulted in a low ethanol product that might be the strategy for the development of health-targeted low alcohol beverages. Several studies indicated the potential health benefits from the production of low alcoholic beverages, such as lowering cholesterol and increasing high-density lipoprotein (HDL), which helps to prevent heart disease and the risks of alcohol-related illnesses [19]. The genus *Cyberlindnera*, as well as *C. subsufficiens*, *C. mrakii*, *C. jadinii*, *C. fabianii* and *C. saturnus*, have been found to produce low-alcoholic beers in both mono and co-fermentation with *S. cerevisiae* [9,38,39,46]. In addition, *W. anomalus* was targeted to be used in alcohol reduction in fermented beverages. Furthermore, a low ethanol producing non-*Saccharomyces* yeast has been also targeted for use in the high level formation of aromatic compounds, such as esters, higher alcohols and fatty acids [44,47]. 

Organic acids in traditional MF-broth residual byproducts have been quantified and include acetic, citric, glucuronic, gallic, oxalic and succinic acids (Table 3), which were similar to the organic acids found in the *Miang* sample [3,48]. These acids were assumed to be released from the *Miang* sample during fermentation. Consistent with the previous reports on tea products, green tea and black tea were the best precursors for organic acids of the kombucha culture, such as acetic acid and glucuronic acid, respectively [49]. 

During fermentation, acetic acid was found as the main organic acid in MF-broth byproduct fermentation followed by succinic, glucuronic, lactic, citric, gallic acid and oxalic acid, respectively. Figure 6 indicates the fluctuation of the total acidity during fermentation. Interestingly, CF fermentation significantly increased the level of acetic acid and glucuronic acid levels (Table 3), which are special organic acids that are reported to be associated with health benefits in kombucha [50]. The increased acetic acid content might be due to the synthesis by yeast metabolically, is derived through the phosphogluconate pathway, acetate kinase pathway and citric acid metabolism [34]. In *S. cerevisiae*, a direct relationship has been established between glycerol and acetic acid production during fermentation [51]. Whereas, *W. anomalus* has been reported to produce a high level of acetic acid, which is one factor that is useful for inhibiting other microbial processes that occur during wine fermentation with *S. cerevisiae* [52]. However, acetic acid is the main acid responsible for the wine fault termed volatile acidity and is a contributor taste ingredient that gives fermented foods their typical strong, pungent and vinegar flavor [44,53]. A glucuronic acid increase detected in MF-broth might be due to the oxidative pathway converting glucose to gluconic acid with yeast [54]. Glucuronic acid is one of the most important components found in kombucha tea that is a vitamin C precursor and plays an important role in the formation of glycosaminoglycans which allows for the detoxifying function via conjugation [55]. Other organic acids found in MF-broth fermentation, including citric, succinic, and oxalic acids are important intermediates in the Krebs cycle. The change in the content of these compounds could be attributed to the yeast cell catabolism [51,52]. According to studies, succinic acid possesses the bitterness and saltiness, and the reduced succinic acid content after fermentation may have the advantages in improving wine taste. 

A PCA was used to analyze the results of products, in terms of glucose, ethanol and organic acids that remained after fermentation (Figure 7). The total variance was 78.09%, PC1 (46.57%) and PC2 (31.52%). Glucose, pH and gallic acids had been represented at PC1, whereas ethanol and organic acids were revealed at PC2. Regarding the relevance of the PC score, the PC1 projects the sample of the original MF-broth (0h), control, P3, and P2 + P3. PC2 was differentiated among samples of TISIR 5088, P2, TISIR 5088 + P2, TISIR 5088 + P3 and TISIR 5088 + P2 + P3. A clear separation of the fermented MF-broth obtained with the *S. cerevisiae* TISTR5088 was strongly related to ethanol production. In contrast, glucose concentration was observed in the unfermented sample of the initial MF-broth and the control at 120 h. The final important product, organic acids correlated with the original MF-broth that might come from what remains of the biotransformation during the FFP *Miang* fermentation, except gallic acid that was most associated with *C. rhodanensis* P3.

### 3.5. Phenolic Compounds and Antioxidant Activity

Phenolic compounds, which are natural antioxidants, have markedly generated interest in *Miang* production due to their potential beneficial effects on humans [3,4,26]. Interestingly, the MF-broth residual byproduct is similar to the extraction content of fermented leaves with water during the anaerobic fermentation, which causes the water-soluble phenolic compounds to be solubilized after release from the *Miang* leaves. Therefore, the fermented MF-broth byproduct was expected to have the same benefits as other types of fermented tea beverages as well. The overall perception of the experimental analysis with the yeast inoculation in fermentation revealed that TP, TT and TF were similar to the initial control. TP values ranged from 6.61 ± 0.28 to 8.27 ± 0.31 g/L, TT values ranged from 6.00 ± 0.02 to 7.91 ± 0.12 g/L and TF values ranged from 0.20 ± 0.03 to 0.36 ± 0.08 g/L (Figure 8). The antioxidant capacity of SF and CF were similar, where 10 µL of each sample exhibited DPPH scavenging abilities at a range of 68.93 ± 1.47% to 71.43 ± 1.7%, which showed no difference between the initial MF-broth and after 120 h fermentation. The bioactive molecules in the broth might be the result of biotransformation in the SMF process, in which microbes convert macronutrients, such as carbohydrates, proteins and lipids into bioactive compounds and dissolve them in aqueous solutions during fermentation [4]. This study added only glucose as a carbon source without other substrates for yeast to grow in and convert into ethanol. As a result, there may be insufficient precursors in MF-broth fermentation to be converted into the bioactive components. However, by comparing the bioactive compounds in the fermented *Miang*, it was found to be close to or higher than other fermented tea beverages. For example, in kombucha made from black, white and red teas, the TP content after 7 days of fermentation, was at 2.19 ± 0.21, 2.05 ± 0.30, and 2.70 ± 0.40 g/L, respectively, whereas the DPPH scavenging ability was 70.63 ± 0.53%, 79.13 ± 0.93% and 77.37 ± 0.80% [56]. It is expected that by using the benefits of tea containing catechins, gallic acid and other antioxidative compounds, the fermented MF-broth could be developed into a health-targeted beverage, similar to other tea wines, such as Oolong tea and pear wine [18].

Tea phenolic compounds were identified and quantified in MF-broth products, including catechins, gallic acid, pyrogallol and caffeine. In the original MF-broth C, GC, EGC, EC and EGCG were confirmed and this was similar to that in *Miang* FFP [3]. However, following the fermentation of MF-broth byproduct substrate, the tea polyphenols were significantly decreased, except EC and EGC (Table 3). Considering the amount of each substance, C increased while GC decreased in all fermentation samples. EC, EGC and EGCG were stable, except for the samples inoculated with *C. rhodanensis* P3, in the single and co-cultured fermentations, and showed a higher content of EC and EGC, compared to the original MF-broth (Table 3). The increase in EC and EGC might be due to the hydrolysis of gallate polyphenols (EGCG) by esterase or tannase from the yeast [36]. However, the reduction of the total catechins in all fermented samples might be from the biotransformation by the yeast. That may lead to the contribution of polyphenols to MF-broth and simultaneously the reduction of the bitter and astringent tastes of MF-broth.

### 3.6. β-Glucosidase Activity in MF-Broth Fermentation

Changes of β-Glucosidase activity in MF-broth were observed and it was found that the increase in enzyme activity was observed in all fermentation broths, except the non-inoculated treatment (control). The SF fermented MF-broth with *S. cerevisiae* TISTR 5088 produced β-glucosidase at 24.51 ± 1.42 to 46.09 ± 1.73 mU/mL, whereas non-*Saccharomyces W. anomalus* P2 and *C. rhodanensis* P3 produced β-glucosidase in the range of 63.47 ± 5.12 to 89.86 ± 5.31 mU/mL and 20.78 ± 1.42 to 196.68 ± 3.55 mU/mL, respectively. *C. rhodanensis* P3 showed the highest β-glucosidase activity of 196.68 ± 3.55 mU/mL at 24 h fermentation. In the case of the CF fermentation, all treatments showed the activity of β-glucosidase at 12 h to 48 h except TISTR 5088 + P3, which was continuous to 120 h (Figure 9). TISTR 5088 + P3 showed the highest activity of 127.56 ± 1.24 mU/mL at 24 h fermentation and this ranged from 23.35 ± 4.09 to 127.56 ± 1.24 mU/mL, followed by TISTR 5088 + P2 (40.84 ± 0.89 to 97.40 ± 4.44 mU/mL), P2 + P3 (17.59 ± 2.67 to 82.94 ± 1.78 mU/mL) and TISTR 5088 + P2 + P3 (1.63 ± 0.31 to 49.64 ± 1.71 mU/mL), respectively. However, the decrease of β-glucosidase levels after 24 h was observed (Figure 5). This might be due to an increase in ethanol levels and high acid, as the previous research found that high acidity, high glucose and high ethanol concentrations affected the function of this enzyme [57]. However, MF-broth fermentation with *C. rhodanensis* P3 showed remarkable enzyme activities in both SF and CF fermentations, in which the functional activity of β-glucosidase is hydrolyzing the glycosidic bonds from various aglycone structures into monoterpenes, which are precursors of aromatic compounds [13,14]. Consequently, previous research investigated the use of β-glucosidase produced from *Cyberlindnera* to develop an aroma in fermented beverages and tea [9,37,39,58]. Wang et al. [58] found that *C. saturuns* var. *mrakii* NCYC 2251 β-glycosides enhanced the tea aroma in green tea slurry by hydrolyzing glycoside precursors (β-glucosides and β-primeverosides) such as geranyl, linalyl, benzyl and 2-phenylethyl glucosides. Furthermore, the benefits of β-glycosides in improving the beverage flavor were also revealed in the *W. anomalus* species, and certain strains were reported to have β-glycosidase, that is multi-functionally active under common oenological conditions. Thus, the enzyme might have multiple applications in fermentation, such as enhancing the sensory and bioactive component concentrations by splitting the glycosylated precursors [6,7]. Due to the reasons mentioned above, β-glucosidase from both non-*Saccharomyces* strains used in this experiment possibly plays an important role in odor and aroma formation, leading to the formation of special quality characteristics in fermented MF-broth byproducts, which requires further research.

### 3.7. Volatile Aromatic Compounds in MF-Broth Fermentation

Figure 10 shows the volatile compounds associated with the metabolic activities of *S. cerevisiae* TISTR 5088, *W. anomalus* P2 and *C. rhodanensis* P3 and their co-culture during fermentation and the original MF-broth from the byproduct. In total, 36 volatile compounds were found in both the unfermented and fermented samples, indicating that alcohol, ester, and terpenes were quantitatively the major group of VOCs in MF-broth byproduct fermentation. A heatmap with hierarchical clustering analysis was used to determine whether the volatile flavor compounds associated with each sample could stimulate the sample clustering, based on the Pearson correlation. Cluster A, which represented the control, had limited effect on the aroma component profile and a significant concentration effect of vitispirane, as shown in Figure 10. The original MF-broth byproduct and yeast-associated fermented samples produced increasing levels of esters, acids and alcohols in Cluster B. Cluster C gathered yeast and co-culturing effects that produced more esters, acids, and alcohols after 120 h of fermentation.

The original MF-broth VOCs were considered the byproduct of SMF *Miang* biotransformation, with microorganisms probably converting the tea aromatic precursors, including carotenoids, lipids, glycosides and other compounds, as previously reported [59,60]. As a result, 22 aromatic substances were detected in the original MF-broth, accounting for 70.16% of the total aroma content. Linalool and methyl salicylate were the main VOCs in that broth, followed by linalool oxide, (Z)-3-hexen-1-ol, benzyl alcohol, ethyl acetate, isoamyl acetate and other substances. Following the preparation of the MF-broth substrate with 5% (*w/v*) glucose, the VOCs profiles changed with linalool, methyl salicylate, ethyl acetate and isoamyl acetate decreasing with the production process. In contrast, new compounds, such as vitispirane, β-ocimene, limonene and α-terpinene, were discovered in the MF-broth substrate. Vitispirane had a high level of VOCs, which has been described as floral, fruity, woody or reminiscent of eucalyptus [61]. Compared to this, the further fermented MF-broth byproduct with yeast was significantly different. Higher levels of flavor intensity (total peak area) were discovered at 120 h fermentation with co-culture, when compared to the unfermented and fermented MF-broth byproducts. This result demonstrated the synergistic effect of co-culturing on the production of aroma compounds during MF-broth fermentation. 

The fermented MF-broth was clearly in the hierarchical clustering that separated the between-group yeast providing alcohols and ester (Figure 10). In the alcohol group, isoamyl alcohol exhibited high levels in all fermentations with *S. cerevisiae* TISTR 5088 and *W. anomalus* P2, in SF and CF. Meanwhile, *C. rhodanensis* P3 represented a high amount of ester group, ethyl acetate and isoamyl acetate, which were remarkable in both of SF and CF. During fermentation, yeast uses the Ehrlich pathway to convert amino acids or sugars into isobutanol, isoamyl alcohol and 2-phenylethanol [34,62,63,64]. Dzialo et al. [62] suggested isoamyl alcohol and 2-phenylethanol were the major fuel alcohols found in alcoholic beverages. In this study, MF-broth fermentation revealed the presence of isoamyl alcohol and 2-phenylethanol, with alcohol produced at a higher level in both the SF and CF with *S. cerevisiae* TISTR 5088 and *W. anomalus* P2, which was consistent with previous studies [34,36,43,65]. However, although in the original MF-broth, a high quantity of (Z)-3-hexen-1-ol and benzyl alcohol (known as tea leaf alcohol) was found, after fermentation that compound was reduced by the yeast [66]. 

Non-*Saccharomyces* yeast strains are known as the efficient producers of esters in fermented foods and beverages. The earlier result of VOCs formed by *W. anomalus* P2 and *C. rhodanensis* P3, the confirmed ethyl acetate (fruity aroma), isoamyl acetate (banana aroma) and 2-phenethyl acetate (2PA), have been described as good characteristics and pleasant aromatic properties to wine [67,68]. In this investigation, the higher ester in fermented MF-broth was generated with *C. rhodanensis* P3, followed by *W. anomalus* P2, in SF and CF. Isoamyl acetate and ethyl acetate were dominant esters found in all fermented MF-broths, which were inoculated with *C. rhodanensis* P3. Furthermore, in *C. rhodanensis* P3, benzyl acetate, 2-phenethyl acetate, furfuryl acetate, ethyl decanoate and neryl acetate were discovered, all of which were not found in the other samples without P3. Different ester characteristics in MF-broth might offer a unique odor in each sample. Several reports confirmed that esters contribute the most to the aroma of alcoholic beverages. The principal contribution to the typical fruity scents of the fermentation aroma is a mixture of ethyl caproate and ethyl caprylate (apple-like aroma), while isoamyl acetate imparts a banana-like aroma and 2-phenylethyl acetate contributes a fruity and flowery flavor [8,69,70]. However, *Cyberlindnera* yeast has been reported to produce high concentrations of acetate esters, in particular isoamyl acetate, ethyl acetate and 2-phenylethyl acetate [37]. In particular, the *Cyberlindnera* species is known for its high ester production, which was shown in the studies with *C. saturnus, C. mrakii* and *C. subsufficiens*. Furthermore, it has been suggested to use yeasts with a high production of flavor compounds (i.e., esters, higher alcohols) to mask the wort-like flavor of non-alcoholic beer, cider, and wine [69,71,72]. This study is the first report on the VOCs’ formation by *C. rhodanensis* in a fermented beverage. 

Terpenes, volatile phenols, and other esterified VOCs were also found in fermented MF-broth. For example, the terpene groups found were β-ocimene, linalool oxide, terpineol and geraniol, whereas the volatile phenols found were benzenol and 4-ethylphenol. In general, terpenes belong to the primary aroma group that contribute to desirable descriptors, such as flowery, honey and citrus notes. Linalool and terpineol have low odor thresholds and offer floral aromas [13,57]. Meanwhile, after 120 h of fermentation, several VOCs, such as methyl salicylate, (Z)-3-hexen-1-ol and linalool, were dramatically reduced. The flavor characteristic of MF-broth increased as fermentation proceeded. Following 120 h of fermentation, more acetate esters and acids were generated but there were fewer terpenoid alcohols than in the initial MF-broth. Terpenoid alcohols can be liberated from glycosides by β-glucosidase or produced via yeast during fermentation, resulting in their accumulation in fermented samples. Meanwhile, terpenoid alcohols can be isomerized into other terpenoids or converted into corresponding acetate esters. For example, methyl salicylate, (Z)-3-hexen-l-ol, 2-phenylethanol and benzyl alcohol were released with glucosides, and the alcohols can be further transferred into esters by yeast [58,73].

The PCA of VOCs evaluated the relationship between VOCs and the fermentation method. The first two principal components (PCs) accounted for 75.03% of the total variance, of which PC1 and PC2 accounted for 55.20% and 19.83%, respectively. The fermented MF-broth was divided into four groups, based on their similarity on the score plot (Figure 11). Group 1 was positioned in the lower right quadrant and was fermented with P3, P2 + P3 and TISTR 5088 + P3, which produced more acetate esters, particularly ethyl acetate, isoamyl acetate, 2-phenylethyl acetate, furfuryl acetate, benzyl acetate, cis-3-hexenyl formate and neryl acetate, and terpenes, such as 2,6-dimethyl-2,6-octadiene, β-ocimene and 3,7-dimethyl-2,6-octadien-1-ol. Meanwhile, TISTR 5088 + P2 and TISTR 5088 + P2 + P3 were located in the upper right quadrant (Group 2) due to the higher levels of octanoic acid, ethyl caprate and N-ethyl acetamide synthesis. The fermented MF-broth with TISTR 5088, P2, and original formed Group 3, corresponded with high alcohols and terpenes, particularly isoamyl alcohol, phenylethyl alcohol, benzenol, 3-octanol linalool, (Z)-3-hexen-l-ol, geraniol, benzyl alcohol, linalool oxide and cis-linalool. The control was located in the lower left quadrant, distant from the sequential fermentations, which were characterized by methyl salicylate, 4-ethylphenol, terpineol, limonene, α-terpinene, allo-ocimene and vitispirane. According to the PCA results, the yeast fermentation had a significant impact on the aroma profile of MF-broth. Therefore, this study indicates the performance of two VOCs groups on the properties of the fermented MF-broth, including the formation of higher alcohols by *S. cerevisiae* TISTR 5088 and *W. anomalus* P2, and higher esters by *C. rhodanensis* P3, and clearly showed the VOCs and correlation of yeast group between unfermented and fermented MF-broth. The results of this experiment demonstrated the ability of non-*Saccharomyces* yeast in the improvement of fermented MF-broth odor.

As aforementioned, although the original MF-broth benefits from partial bioactivity during the *Miang* fermentation processes, such as organic acids, catechins and other substances. The original MF-broth, Moreover, has a disagreeable odor for consumption. As a result, it was interesting to search for opportunities to strengthen the flavor while also providing additional benefits. In this research, we selected non-*Saccharomyces* that exhibited a distinct odor for the MF-broth to enhance the scent and add bioactive compounds, as well as the advantages from a low-alcohol beverage. We expect this study will be advantageous in the development of healthy drinks and add value to the MF-broth. Future iterations of this research will be the sensory evaluation for consumer acceptance and commercial development.

## 4. Conclusions

The findings of this study demonstrated the potential for using the waste byproduct as MF-broth residual byproduct substrate in the development of health-targeted beverages using non-*Saccharomyces* yeast species. Organic acids that were important in fermentation include acetic, citric, glucuronic, lactic, succinic, oxalic and gallic acids. The bioactive compounds and antioxidant activity were similar with the initial fermentation stage. β-Glucosidase was found to be involved in the quality improvement of fermented MF-broth by enhancing the volatile alcohols and esters compounds. Thirty-six volatile compounds were detected in both the unfermented and fermented samples, indicating that alcohol, ester, and terpenes were the most abundant VOCs in the MF-broth byproduct fermentation. The PCA and hierarchical clustering analysis clearly showed the VOCs and the correlation of the yeast group that classified Cluster A (control), Cluster B (original MF-broth) and Cluster C evaluated the yeast and co-culturing that created more esters, acids, and alcohols.

## Figures and Tables

**Figure 1 jof-09-00165-f001:**
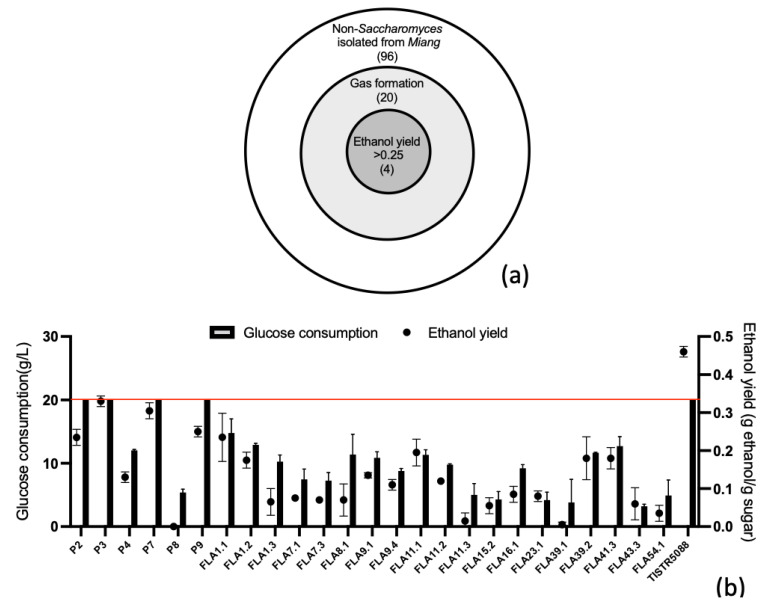
Screening for the ethanol producing yeast isolated from *Miang*. The graphical ratio of a total screened yeast number, in comparison to the gas-forming and ethanol-producing isolates (**a**). Glucose consumption (g/L) and ethanol yield (g/g sugar consumption) of 24 yeast isolates and *S. cerevisiae* TISTR 5088 (control) after 3 days of fermentation in YPD medium at 30 °C (**b**).

**Figure 2 jof-09-00165-f002:**

Hemolytic activity of yeast isolates P2, P3, P7 and P9 on blood agar at 30 °C for 7 days (**a**) and growth of the yeast isolate on YMA containing 50 g/L tannin at 37 °C for 3 days (**b**).

**Figure 3 jof-09-00165-f003:**
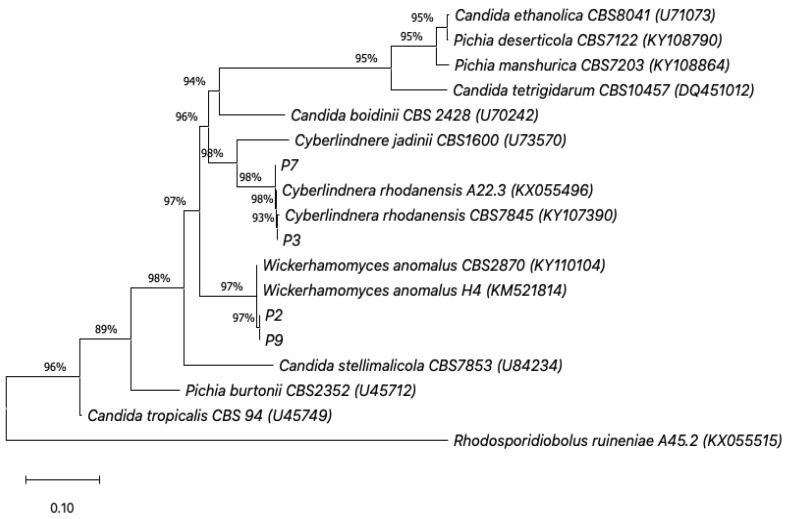
Phylogenetic tree analysis of the yeast *Cyberlindera rhodanensis* P3 and P7, *Wikerhamomyces anomalus* P2 and P9, based on particle 26S rDNA D1/D2 region sequence using the neighbor-joining method. Bootstrap values >50% (based on 1000 replication) are given at the branch points. The scale bar shows a patristic distance of 0.10.

**Figure 4 jof-09-00165-f004:**
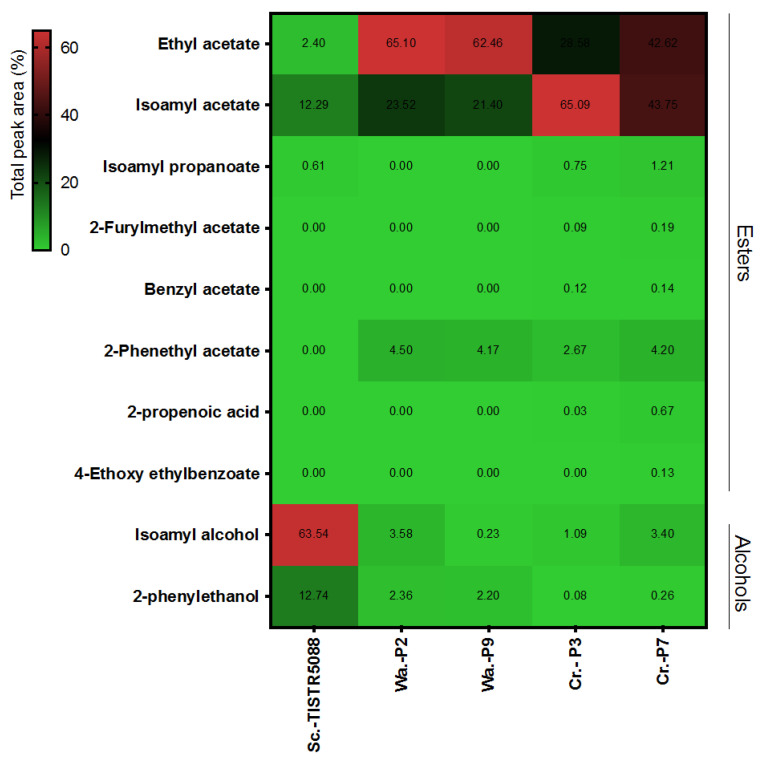
Heatmap using the total peak area (%) of VOCs produced with *S. cerevisiae* TISTR 5088, *W. anomalus* (P2, P9) and *C. rhodanensis* (P3, P7) on YM agar at 30 °C for 2 days.

**Figure 5 jof-09-00165-f005:**
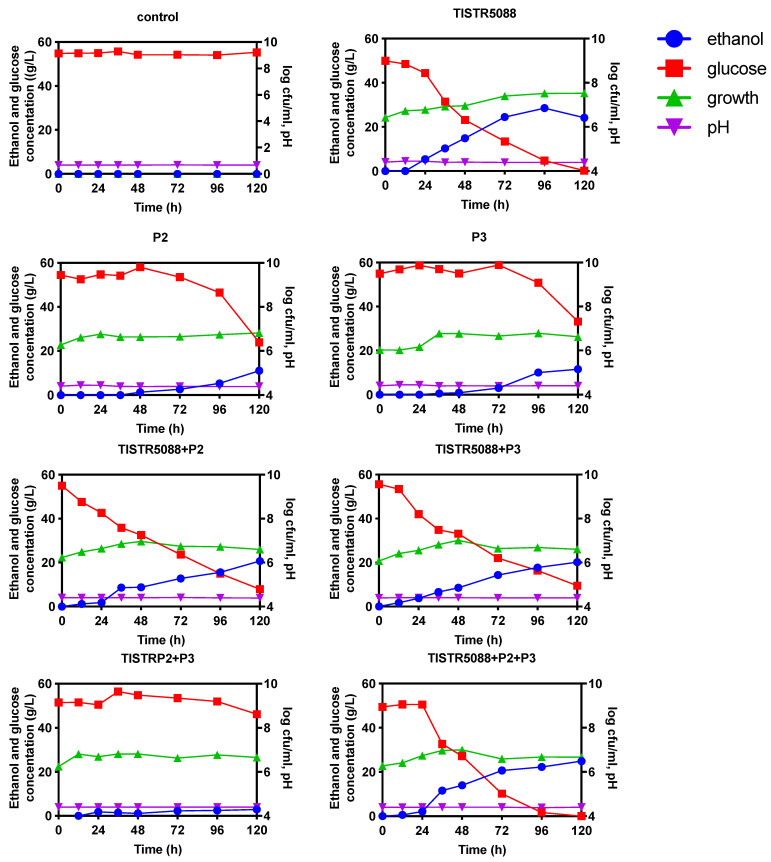
Cell growth, glucose, and ethanol concentration profiles during the fermentation of MF-broth fermented by *S. cerevisiae* TISTR 5088, *W. anomalus* P2 and *C. rhodanensis* P3, with single and co-fermentation at 30 °C for 120 h.

**Figure 6 jof-09-00165-f006:**
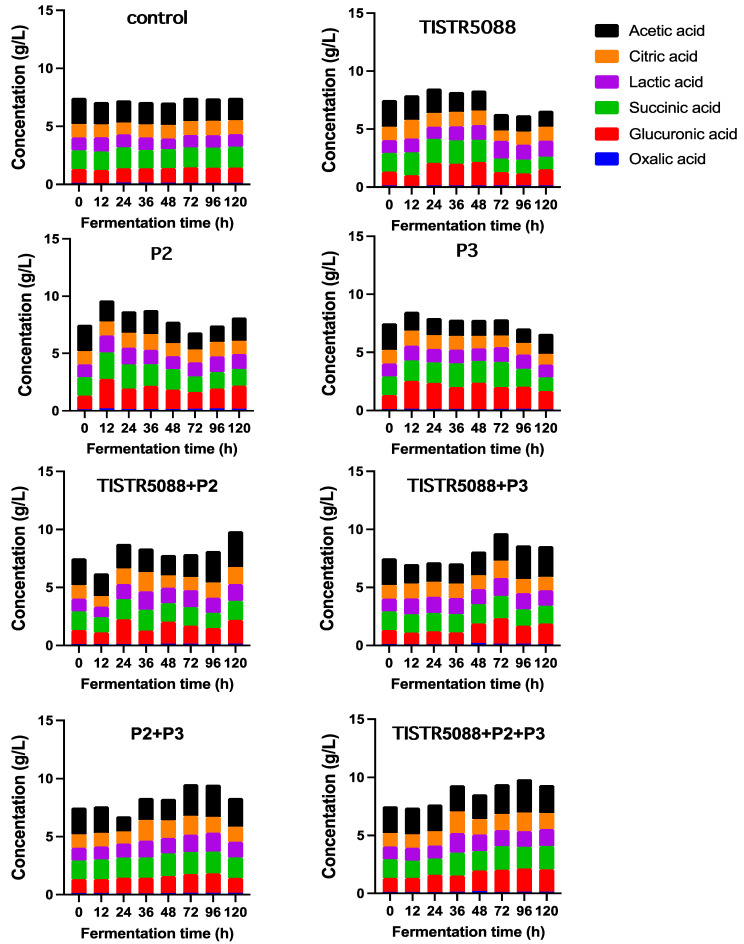
Organic acid profiles during fermentation of MF-broth fermented by *S. cerevisiae* TISTR 5088, *W. anomalus* P2 and *C. rhodanensis* P3, with single and co-fermentation at 30 °C for 120 h.

**Figure 7 jof-09-00165-f007:**
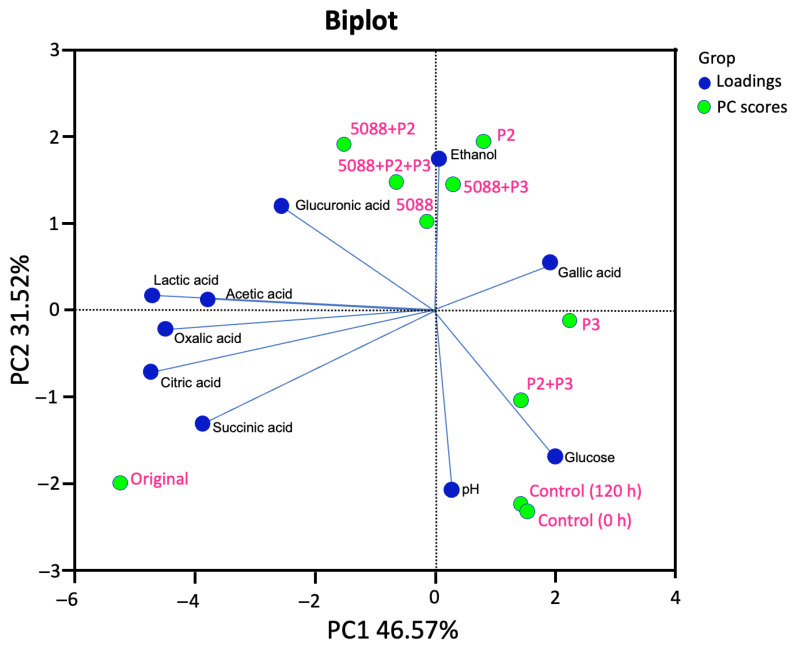
PCA biplot of the organic acids, glucose, ethanol, and pH variables in the original and fermented MF-broth by *S. cerevisiae* TISTR 5088, *W. anomalus* P2 and *C. rhodanensis* P3, with single and co-fermentation at 30 °C for 120 h.

**Figure 8 jof-09-00165-f008:**
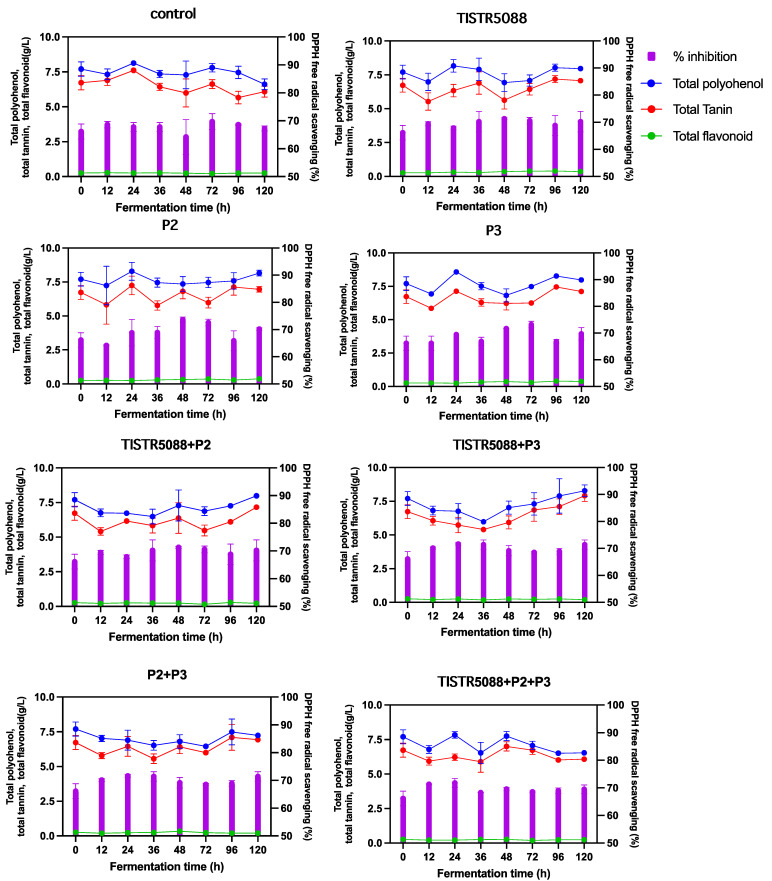
Bioactive compound and antioxidant profiles during the fermentation of MF-broth fermented by *S. cerevisiae* TISTR 5088, *W. anomalus* P2 and *C. rhodanensis* P3, with single and co-fermentation.

**Figure 9 jof-09-00165-f009:**
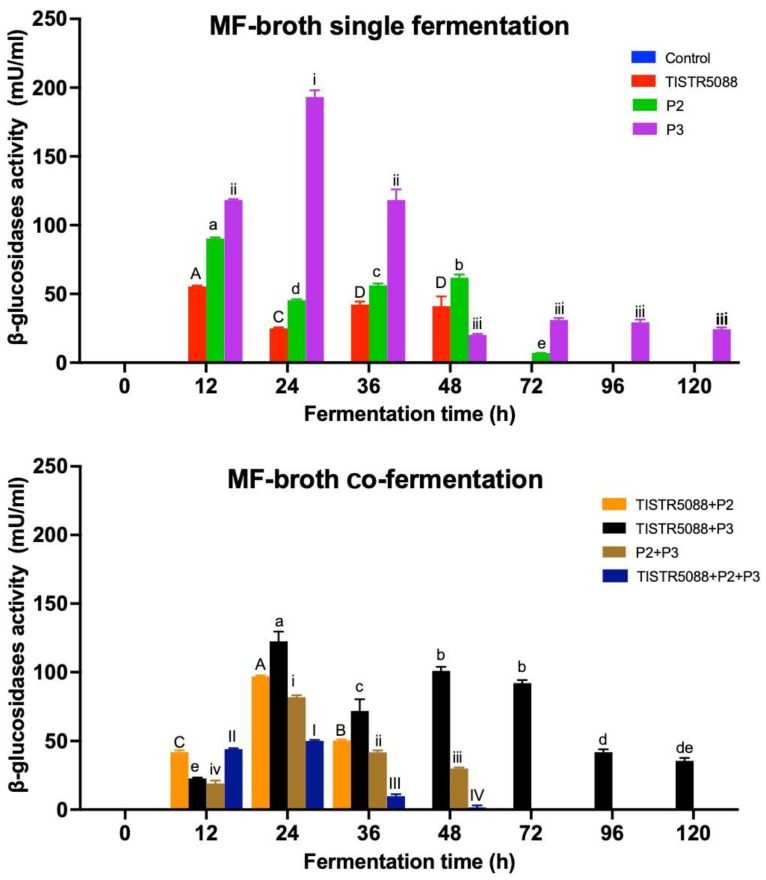
β-glucosidase activity of non-inoculated (control), *S. cerevisiae* TISTR 5088, *W. anomalus* P2 and *C. rhodanensis* P3 during MF- broth fermentation.

**Figure 10 jof-09-00165-f010:**
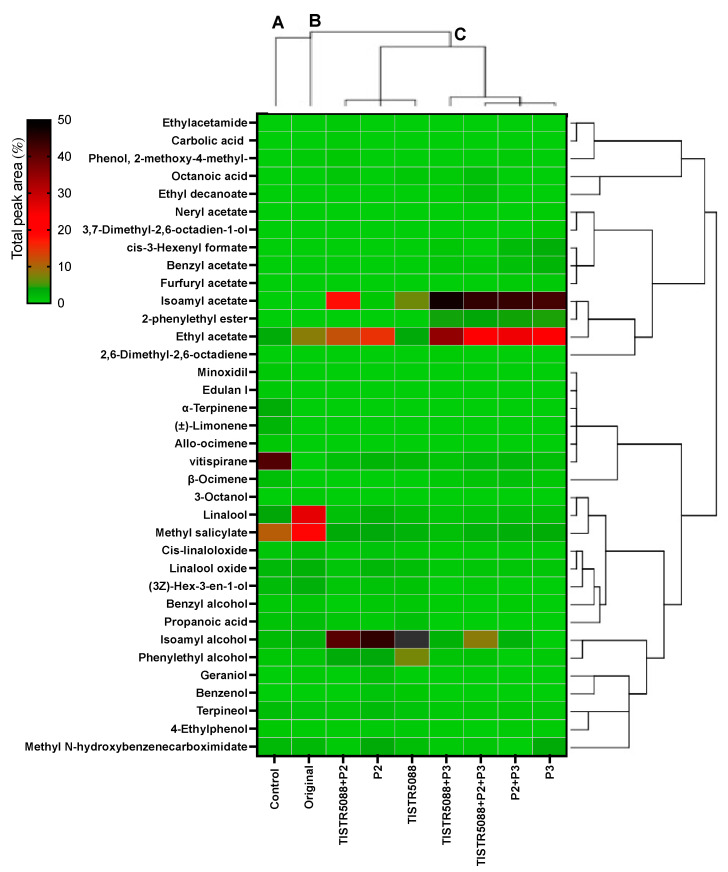
Heatmap and hierarchical cluster analysis representation corresponding to the 36 volatile compounds of original and fermented MF-broth inoculated with *S. cerevisiae* TISTR 5088, *W. anomalus* P2 and *C. rhodanensis* P3 at 30 °C for 120 h.

**Figure 11 jof-09-00165-f011:**
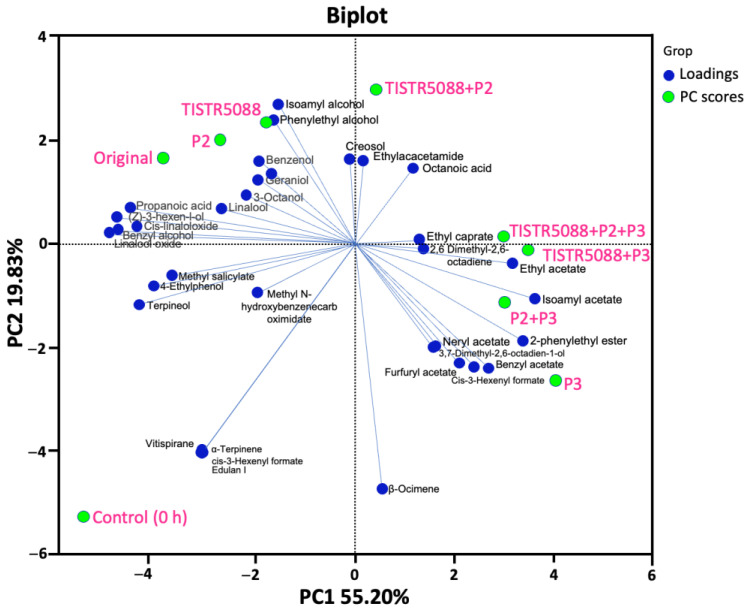
PCA biplot of the volatile compounds in the original and fermented MF-broth by *S. cerevisiae* TISTR 5088, *W. anomalus* P2 and *C. rhodanensis* P3, with single and co-fermentation at 120 h.

**Table 1 jof-09-00165-t001:** Probiotic characteristics of four selected yeast strains.

Isolate	Survival Rate (%)			Microbial Adhesion to Solvents (%)
	Bile Salt	pH2	pH3	
*W. anomalus* P2	91.42 ± 0.10 ^a^	91.76 ± 0.18 ^b^	96.40 ± 0.08 ^b^	43.81 ± 0.89 ^c^
*C. rhodanensis* P3	70.28 ± 0.81 ^d^	83.88 ± 0.15 ^c^	99.71 ± 0.09 ^a^	39.45 ± 0.54 ^d^
*W. anomalus* P7	66.61 ± 0.68 ^e^	84.67 ± 0.11 ^c^	99.94 ± 0.08 ^a^	31.14 ± 0.62 ^e^
*C. rhodanensis* P9	75.51 ± 0.12 ^c^	91.86 ± 0.23 ^b^	95.21 ± 0.12 ^b^	52.31 ± 0.94 ^b^
*S. cerevisiae* TISTR 5088	85.59 ± 0.66 ^b^	95.93 ± 0.01 ^a^	99.61 ± 0.01 ^a^	63.77 ± 1.02 ^a^

Note: ^a–e^ Different letters within the same column indicate a statistically significant difference (*p* < 0.05).

**Table 2 jof-09-00165-t002:** Carbon fermentation and assimilation characteristics of the four selected yeast strains.

Characteristic	Strains				Characteristic	Strains			
	P2	P3	P7	P9		P2	P3	P7	P9
Assimilation of Carbon					Assimilation of Carbon				
D-galactose	+	-	-	+	Potassium gluconate	-	w	w	-
Cyacloheximide (actidione)	-	-	-	-	Levulinic acid	-	-	-	-
D-saccharose (sucrose)	+	+	+	+	D-Glucose	+	+	+	+
N-Acetyl-glucosamine	+	-	-	+	L-Sorbose	-	w	w	-
Lactic acid	+	+	+	+	Glucosamine	-	-	-	-
L-arabinose	-	-	-	-	Esculin	+	+	+	+
D-Cellobiose	-	+	+	-	**Fermentation**				
D-Raffinose	+	-	-	+	Cellobiose	-	+	+	-
D-Maltose	+	+	+	+	D-glucose	+	+	+	+
D-Trehalose	-	+	+	-	D-galactose	+	-	-	+
Potassium 2 ketogluconate	-	-	-	-	Maltose	+	w	w	+
Methyl-αD-glucopyranoside	+	-	-	+	Sucrose	+	+	+	+
D-Mannitol	+	-	-	+	Lactose	-	-	-	-
D-Lactose (bovine origin)	+	-	-	+	Trehalose	-	-	-	-
Inositol	-	-	-	-	Raffinose	+	-	-	+
D-Sorbitol	+	+	+	+	Xylose	-	+	+	-
D-Xylose	+	+	+	+	**Other Growth Characteristics**				
D-Ribose	+	-	-	+	50% glucose yeast extract	+	+	+	+
Glycerol	+	+	+	+	60% glucose yeast extract	+	+	+	+
L-Rhamnose	-	+	+	-	10% NaCl	-	-	-	-
Palatinose	+	+	+	+	Growth at 20 °C	+	+	+	+
Erythritol	+	-	-	+	Growth at 25 °C	+	+	+	+
D-Melibiose	-	-	-	-	Growth at 30 °C	+	+	+	+
Sodium glucuronate	-	-	-	-	Growth at 37 °C	+	+	+	+
D-Melezitose	+	+	+	+	Growth at 45 °C	-	-	-	-

Note: +, positive; -, negative; w, weak positive.

**Table 3 jof-09-00165-t003:** Physicochemical properties of fermented *Miang* broth with single and co-cultures of *S. cerevisiae* TISTR 5088, *W. anomalus* P2 and *C. rhodanensis* P3 at 120 h after fermentation.

Property	*Miang* Broth Original	Control (0 h)	Following 120 h Fermentation							
			Control	5088	P2	P3	5088 + P2	5088 + P3	P2 + P3	5088 + P2 + P3
Glucose (g/L)	3.56 ± 0.02	53.21 ± 0.02 ^h^	53.27 ± 0.12 ^h^	0.19 ± 0.01 ^b^	23.86 ± 0.45 ^e^	33.19 ± 0.32 ^f^	7.93 ± 0.12 ^c^	9.50 ± 0.21 ^d^	46.20 ± 1.09 ^g^	0.00 ± 0.00 ^a^
Ethanol (g/L)	0.00 ± 0.00 ^f^	0.00 ± 0.00 ^f^	0.00 ± 0.00 ^f^	23.67 ± 0.70 ^b^	11.03 ± 0.07 ^d^	11.56 ± 0.00 ^d^	20.74 ± 0.0 ^c^	20.15 ± 0.50 ^c^	2.93 ± 0.0 ^e^	24.91 ± 0.0 ^a^
pH	4.06 ± 0.0 ^a^	4.1 ± 0.0 ^a^	4.08 ± 0.03 ^a^	3.9 ± 0.04 ^a^	3.94 ± 0.00 ^a^	4.04 ± 0.06 ^a^	3.91 ± 0.03 ^a^	3.97 ± 0.0 ^a^	4.02 ± 0.02 ^a^	4.02 ± 0.0 ^a^
**Organic acid**										
Acetic acid (g/L)	3.04 ± 0.15 ^a^	2.24 ± 0.12 ^d^	1.91 ± 001 ^f^	1.37 ± 0.15 ^f^	2.03 ± 0.12 ^e^	1.72 ± 0.02 ^f^	3.09 ± 0.34 ^a^	2.64 ± 0.25 ^b^	1.48 ± 0.12 ^g^	2.42 ± 0.18 ^d^
Citric acid (g/L)	2.79 ± 0.05 ^a^	1.17 ± 0.09 ^de^	1.22 ± 0.2 ^de^	1.21 ± 0.05 ^de^	1.16 ± 0.05 ^f^	0.93 ± 0.01 ^f^	1.47 ± 0.01 ^b^	1.17 ± 0.02 ^de^	1.28 ± 0.03 ^cd^	1.38 ± 0.02 ^bc^
Glucuronic acid (g/L)	1.70 ± 0.15 ^b^	1.20 ± 0.01 ^e^	1.31 ± 0.2 ^de^	1.39 ± 0.03 ^d^	2.02 ± 0.23 ^a^	1.57 ± 0.04 ^c^	2.04 ± 0.04 ^a^	1.75 ± 0.02 ^b^	1.29 ± 0.01 ^de^	1.94 ± 0.01 ^a^
Lactic acid (g/L)	1.90 ± 0.01 ^a^	1.10 ± 0.03 ^d^	1.08 ± 0.05 ^d^	1.38 ± 0.03 ^bc^	1.31 ± 0.01 ^c^	1.08 ± 0.02 ^d^	1.44 ± 0.05 ^b^	1.31 ± 0.01 ^bc^	1.37 ± 0.03 ^bc^	1.45 ± 0.03 ^b^
Oxalic acid (g/L)	0.22 ± 0.02 ^a^	0.13 ± 0.02 ^bc^	0.13 ± 0.01 ^bc^	0.15 ± 0.01 ^bc^	0.17 ± 0.04 ^ab^	0.10 ± 0.01 ^c^	0.16 ± 0.01 ^bc^	0.13 ± 0.03 ^bc^	0.15 ± 0.01 ^bc^	0.14 ± 0.01 ^bc^
Succinic acid (g/L)	2.59 ± 0.06 ^a^	1.61 ± 0.05 ^de^	1.81 ± 0.14 ^c^	1.08 ± 0.07 ^g^	1.45 ± 0.01 ^f^	1.19 ± 0.01 ^g^	1.65 ± 0.04 ^de^	1.55 ± -0.05 ^ef^	1.76 ± 0.05 ^cd^	2.02 ± 0.01 ^b^
**Bioactive compounds**										
Total polyphenol (g/L)	7.86 ± 0.2 ^a^	7.70 ± 0.98 ^a^	6.61 ± 0.28 ^b^	7.97 ± 0.09 ^a^	8.15 ± 0.015 ^a^	7.98 ± 0.06 ^a^	7.99 ± 0.12 ^a^	8.27 ± 0.31 ^a^	7.24 ± 0.02 ^b^	6.54 ± 0.07 ^b^
Total tannin (g/L)	6.26 ± 0.0 ^b^	6.73 ± 0.21 ^ab^	6.09 ± 0.35 ^b^	7.07 ± 0.28 ^ab^	6.96 ± 0.09 ^ab^	7.12 ± 0.15 ^ab^	7.71 ± 0.06 ^a^	7.91 ± 0.12 ^a^	6.94 ± 0.31 ^ab^	6.00 ± 0.02 ^ab^
Total flavonoid (g/L)	0.30 ± 0.01 ^a^	0.27 ± 0.09 ^a^	0.26 ± 0.23 ^a^	0.36 ± 0.08 ^a^	0.37 ± 0.13 ^a^	0.37 ± 0.07 ^a^	0.23 ± 0.01 ^a^	0.20 ± 0.02 ^a^	0.20 ± 0.03 ^a^	0.24 ± 0.01 ^a^
DPPH free radical scavenging (%)	67.91 ± 0.06 ^a^	66.15 ± 2.69 ^a^	67.28 ± 0.90 ^a^	68.93 ± 1.47 ^a^	71.38 ± 1.22 ^a^	69.69 ± 2.35 ^a^	70.23 ± 3.78 ^a^	71.43 ± 1.70 ^a^	69.49 ± 1.60 ^a^	70.23 ± 1.13 ^a^
Total catechin (g/L)	2.22 ± 0.01 ^a^	1.4 ± 0.02 ^d^	1.24 ± 0.01 ^e^	1.22 ± 0.01 ^e^	1.23 ± 0.01 ^e^	1.64 ± 0.00 ^b^	1.43 ± 0.00 ^d^	1.53 ± 0.01 ^d^	1.64 ± 0.00 ^b^	1.60 ± 0.01 ^b^
Catechin (g/L)	1.02 ± 0.01 ^a^	0.53 ± 0.02 ^f^	0.51 ± 0.01 ^f^	0.56 ± 0.03 ^e^	0.56 ± 0.03 ^e^	0.77 ± 0.02 ^b^	0.66 ± 0.03 ^d^	0.71 ± 0.12 ^c^	0.75 ± 0.03 ^b^	0.71 ± 0.02 ^c^
Gallocatechin (g/L)	0.74 ± 0.03 ^a^	0.44 ± 0.03 ^b^	0.35 ± 0.02 ^c^	0.25 ± 0.03 ^e^	0.24 ± 0.04 ^e^	0.33 ± 0.03 ^c^	0.26 ± 0.01 ^e^	0.26 ± 0.03 ^e^	0.30 ± 0.01 ^d^	0.30 ± 0.00 ^d^
Epigallocatechin (g/L)	0.13 ± 0.01 ^f^	0.17 ± 0.00 ^f^	0.13 ± 0.01 ^f^	0.15 ± 0.01 ^e^	0.16 ± 0.00 ^de^	0.20 ± 0.01 ^c^	0.19 ± 0.01 ^c^	0.21 ± 0.01 ^b^	0.21 ± 0.02 ^b^	0.26 ± 0.01 ^a^
Epicatechin (g/L)	0.15 ± 0.01 ^d^	0.20 ± 0.00 ^c^	0.20 ± 0.01 ^c^	0.21 ± 0.01 ^c^	0.21 ± 0.01 ^c^	0.26 ± 0.01 ^b^	0.24 ± 0.01 ^b^	0.25 ± 0.01 ^b^	0.29 ± 0.01 ^a^	0.24 ± 0.01 ^b^
Epigallocatechin gallate (g/L)	0.18 ± 0.01 ^a^	0.06 ± 0.001 ^de^	0.05 ± 0.00 ^e^	0.05 ± 0.00 ^e^	0.06 ± 0.01 ^de^	0.08 ± 0.00 ^cd^	0.08 ± 0.02 ^bc^	0.10 ± 0.02 ^b^	0.09 ± 0.01 ^bc^	0.09 ± 0.00 ^bc^
Gallic acid (g/L)	0.13 ± 0.01 ^a^	0.05 ± 0.01 ^b^	0.05 ± 0.01 ^b^	0.06 ± 0.01 ^b^	0.06 ± 0.01 ^b^	0.06 ± 0.00 ^b^	0.05 ± 0.00 ^b^	0.06 ± 0.00 ^b^	0.06 ± 0.00 ^b^	0.06 ± 0.00 ^b^
Pyrogallol (g/L)	0.55 ± 0.01 ^a^	0.32 ± 0.01 ^c^	0.42 ± 0.01 ^b^	0.33 ± 0.02 ^c^	0.31 ± 0.01 ^de^	0.32 ± 0.02 ^cd^	0.29 ± 0.02 ^de^	0.28 ± 0.01 ^e^	0.31 ± 0.01 ^cd^	0.32 ± 0.01 ^cd^
Caffeine (g/L)	0.86 ± 0.02 ^a^	0.55 ± 0.02 ^ef^	0.58 ± 0.02 ^cd^	0.59 ± 0.01 ^bc^	0.58 ± 0.02 ^cd^	0.61 ± 0.02 ^b^	0.53 ± 0.01 ^f^	0.58 ± 0.02 ^cd^	0.57 ± 0.02 ^cd^	0.55 ± 0.03 ^de^

Note: ^a–g^ Different letters within the same row indicate a statistically significant difference (*p* < 0.05).

## Data Availability

Not applicable.

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
