# Peer review of "Assessment of Tannin Tolerant Non-Saccharomyces Yeasts Isolated from Miang for Production of Health-Targeted Beverage Using Miang Processing Byproducts"

_jof, 2023, doi:10.3390/jof9020165_

Round 1

Reviewer 1 Report

your review report for research articles:

Assessment of Tannin Tolerant Non-Saccharomyces Yeasts Isolated from Miang for Production of Health-Targeted Beverage Using Miang Processing Byproducts

Is the manuscript clear, relevant for the field and presented in a well-structured manner?

the manuscript is well structured and the content is clear

Are the cited references mostly recent publications (within the last 5 years) and relevant? Does it include an excessive number of self-citations?

Yes, it includes some self-citations.

Nr.11, 26, 48 (first author)

Nr.9, 10, 11, 12,13, 26, 33, 48 (last author)

Is the manuscript scientifically sound and is the experimental design appropriate to test the hypothesis?

yes, it is

Are the manuscript’s results reproducible based on the details given in the methods section?

yes, it is.

Are the figures/tables/images/schemes appropriate? Do they properly show the data? Are they easy to interpret and understand? Is the data interpreted appropriately and consistently throughout the manuscript? Please include details regarding the statistical analysis or data acquired from specific databases.

Figure 1 a, b , 2,3,4 , 5, 6, 7, 8 and  table 1, 2, 3 are easy to understand.

Figure 9 has to be overworked because the figure is understandable without a color representation and please insert MF (Miang fermentation)

Figure 10 is it really necessary to put in an hierarchical cluster analysis?

Are the conclusions consistent with the evidence and arguments presented?

yes

Please evaluate the ethics statements and data availability statements to ensure they are adequate.

They are appropriate.

Novelty: Is the question original and well-defined? Do the results provide an advancement of the current knowledge?

It is a description and characterization of yeasts for Miang fermentation byproducts.

Suggestions for improvement

It is better to start the introduction with line 71  and to integrate the content of line 48 -70 behind.

line 211, 435 Please use MF (Miang fermentation) broth. It is easier to read and to understand.

line 141 YPD (?)

line 150 HPLC (?)

line 157 PBS (?)

line 158 YM (?)

line 165 OD (?)

line  168 MATS (?)

line 275, 277 1,8ml DI water ? dd water ?

3.7 is a hierarchical cluster analysis necessary?  The statement is clear without it.

 line 772 the text cuts off abruptly, please let it fade away and answer the question if there is a potential or advantage for using non –S.c. in Miang processing byproducts

Author Response

Response to Reviewer 1 Comments

We have revised the manuscript following all suggested points of reviewer and all changed points were highlighted with yellow color in R1 version. Regarding the question from reviewers, the answers/explain are responded one by one as following;

Point 1: It is better to start the introduction with line 71  and to integrate the content of line 48 -70 behind.

Response 1: The introduction part was revised as suggested (line 49-89).

Point 2: line 211, 435 Please use MF (Miang fermentation) broth. It is easier to read and to understand.

line 141 YPD (?)

line 150 HPLC (?)

line 157 PBS (?)

line 158 YM (?)

line 165 OD (?)

line  168 MATS (?)

line 275, 277 1,8ml DI water ? dd water ?

Response 2: All full name were added as mentioned.

Point 3: 3.7 is a hierarchical cluster analysis necessary?  The statement is clear without it.

Response 3: We believed that it is necessary due to the high number of MF-broth VOCs. Without a hierarchical cluster analysis, the difference between control, original MF-broth and yeast and co-culturing was not clearly observed. In addition, line 683-688 demonstrated that the fermented MF-broth was clearly in the hierarchical clustering that distinguished between-group yeast supplied alcohols and esters.

Point 4: line 772 the text cuts off abruptly, please let it fade away and answer the question if there is a potential or advantage for using non –S.c. in Miang processing byproducts

Response 4: We have added more information as suggested (line 775-784).

Reviewer 2 Report

The reviewed material has been prepared very carefully. It contains a lot of results, but they are presented in a clear and transparent way. The description of the methodology is concise but unambiguous. The purpose of the work has been clearly formulated and the conclusion is a response to the stated goal.

Continuing the research, it is worth for the authors to carry out a sensory assessment to show that the potential for the health-promoting properties of the drink is also associated with consumer acceptance.

Below are some detailed comments

line 452. S. cerevisiae should be italic

Subchapter 2.10. Write down what constituted the control sample

Figure 9. The authors did not provide 2.10, which is a control. It seems un-inoculated sterile broth was used as a control. Therefore, the β-glucosidase activity is 0. This should be addressed in the figure caption or legend and in the discussion of the results. Currently, when analyzing Fig. 9, one may get the impression that the results for the control sample are not given (blue in the legend).

Author Response

Response to Reviewer 2 Comments

We have revised the manuscript following all suggested points of reviewer and all changed points will be highlighted with green color in R1 version. Regarding the question from reviewers, the answers or explain are responded one by one as following;

Point 1: line 452. S. cerevisiae should be italic

Response 1: S. cerevisiae was italic as suggested (line 452).

Point 2: Subchapter 2.10. Write down what constituted the control sample

Response 2: An un-inoculated sterile broth was used as a control which is already mentioned at subchapter 2.6 (line 227).

Point 3: Figure 9. The authors did not provide 2.10, which is a control. It seems un-inoculated sterile broth was used as a control. Therefore, the β-glucosidase activity is 0. This should be addressed in the figure caption or legend and in the discussion of the results. Currently, when analyzing Fig. 9, one may get the impression that the results for the control sample are not given (blue in the legend).

Response 3: An un-inoculated sterile broth (control) was added in Figure 9 legend and mentioned in the result as suggested (line 626 and 660).